# ICG-001, an Inhibitor of the β-Catenin and cAMP Response Element-Binding Protein Dependent Gene Transcription, Decreases Proliferation but Enhances Migration of Osteosarcoma Cells

**DOI:** 10.3390/ph14050421

**Published:** 2021-05-01

**Authors:** Geoffroy Danieau, Sarah Morice, Sarah Renault, Régis Brion, Kevin Biteau, Jérôme Amiaud, Marie Cadé, Dominique Heymann, Frédéric Lézot, Franck Verrecchia, Françoise Rédini, Bénédicte Brounais-Le Royer

**Affiliations:** 1Phy-OS, Sarcomes Osseux et Remodelage des Tissus Calcifiés, INSERM, UMR1238, Université de Nantes, 44035 Nantes, France; geoffroy.danieau@univ-nantes.fr (G.D.); sarah.morice@univ-nantes.fr (S.M.); sarah.renault@univ-nantes.fr (S.R.); regis.brion@univ-nantes.fr (R.B.); kevin.biteau@univ-nantes.fr (K.B.); jerome.amiaud@univ-nantes.fr (J.A.); frederic.lezot@univ-nantes.fr (F.L.); franck.verrecchia@univ-nantes.fr (F.V.); francoise.redini@univ-nantes.fr (F.R.); 2Centre Hospitalier Universitaire, Université de Nantes, 44035 Nantes, France; 3Equipe Apoptose et Progression Tumorale, Centre de Recherche en Cancérologie et Immunologie Nantes Angers, CRCINA, INSERM, UMR1232, Université de Nantes, Université d’Angers, 44035 Nantes, France; marie.cade@univ-nantes.fr (M.C.); dominique.heymann@univ-nantes.fr (D.H.); 4LaBCT, Institut de Cancérologie de l’Ouest, 44800 Saint Herblain, France; 5European Associated Laboratory Sarcoma Research Unit, INSERM, University of Sheffield, Sheffield S10 2TN, UK

**Keywords:** osteosarcoma, β-catenin, proliferation, migration, ICG-001

## Abstract

High-grade osteosarcomas are the most frequent malignant bone tumors in the pediatric population, with 150 patients diagnosed every year in France. Osteosarcomas are associated with low survival rates for high risk patients (metastatic and relapsed diseases). Knowing that the canonical Wnt signaling pathway (Wnt/β-catenin) plays a complex but a key role in primary and metastatic development of osteosarcoma, the aim of this work was to analyze the effects of ICG-001, a CBP/β-catenin inhibitor blocking the β-catenin dependent gene transcription, in three human osteosarcoma cell lines (KHOS, MG63 and 143B). The cell proliferation and migration were first evaluated in vitro after ICG-001 treatment. Secondly, a mouse model of osteosarcoma was used to establish the in vivo biological effect of ICG-001 on osteosarcoma growth and metastatic dissemination. In vitro, ICG-001 treatment strongly inhibits osteosarcoma cell proliferation through a cell cycle blockade in the G0/G1 phase, but surprisingly, increases cell migration of the three cell lines. Moreover, ICG-001 does not modulate tumor growth in the osteosarcoma mouse model but, rather significantly increases the metastatic dissemination to lungs. Taken together, these results highlight, despite an anti-proliferative effect, a deleterious pro-migratory role of ICG-001 in osteosarcoma.

## 1. Introduction

High-grade osteosarcomas are the most frequent malignant bone tumors in the pediatric population, with 150 patients diagnosed every year in France [1]. Osteosarcoma affects mainly children and adolescents with an incidence around 18 years. Standard therapeutic approaches include conservative surgery associated with poly-chemotherapy. However, patient survival has not evolved for the past decades and remains closely related to the response of tumor cells to chemotherapy, reaching 70% at 5 years for patients with localized forms while less than 30% in metastatic diseases and patients resistant to chemotherapy [2,3,4,5].

The Wnt proteins constitute a large family of secreted glycoproteins that bind to receptors encoded by the frizzled (Fzd) genes. The complexity of the Wnt signaling pathways relies on the fact that 19 Wnt ligands and 10 Fzd receptors have been identified with transduction of their signals through both canonical and non-canonical pathways [6]. The canonical Wnt/β-catenin signaling pathway has been well characterized. In the absence of Wnt ligands, β-catenin is phosphorylated by Casein Kinase 1α (CK1α) and Glycogen Synthase Kinase 3β (GSK3β), both present in a destruction complex that also includes Adenomatous Polyposis Coli (APC) and Axin-2, which leads to β-catenin proteasomal degradation [7]. Following the interaction of Wnt ligands with Fzd receptors that forms a complex with low-density lipoprotein receptor related protein (LRP)-5/6 co-receptor, unphosphorylated β-catenin accumulates and translocates into the nucleus. Then, β-catenin associates to members of the T cell factor/lymphoid enhancer factor (TCF/LEF) family and transcriptional co-activators including cAMP response element-binding protein (CBP) or p300 in order to induce the transcription of various target genes such as CYCLIN D1 or c-MYC [8].

The Wnt signaling pathways regulate important cellular processes including cell proliferation, differentiation or migration and have been involved in many forms of human cancers with complex roles, as promotion or inhibition of tumor initiation, growth or metastasis depending on cancer-stage and type [9,10]. Some studies have focused on the role of Wnt signaling pathways in osteosarcoma development, but current knowledge remains controversial. Indeed, aberrant activation of the canonical Wnt signaling pathway has been highlighted in osteosarcoma cells and seems to be responsible for increased tumorigenicity and proliferation of these cells as well as their metastatic dissemination [11,12]. Thus, high β-catenin levels have been recently reported in osteosarcoma tissues compared to adjacent healthy ones associated with poor prognosis and occurrence of lung metastases [13,14]. However, other studies demonstrated that inactivation of the Wnt/β-catenin pathway plays a key role in osteosarcoma development. In particular, frequent deletions of genes involved in the Wnt signaling pathway have been described in osteosarcoma patients [15,16]. These different data highlight the complexity of the Wnt pathway regulation during primary and metastatic development of osteosarcoma, which constitutes a substantial obstacle to therapeutic targeting of this pathway. Nevertheless, most of the data seems to sustain a pro-tumoral role of the Wnt/β-catenin signaling pathway during osteosarcoma development allowing to hypothesize that inhibiting β-catenin activity should represent a therapeutic strategy in osteosarcoma.

In this context, the aim of the present work was to determine the implication of the β-catenin dependent gene transcription in the primary and metastatic development of osteosarcoma. For this purpose, osteosarcoma cells were treated with ICG-001, a CBP/β-catenin inhibitor blocking the β-catenin dependent gene transcription. On the one hand, ICG-001 has been shown to inhibit tumor development in several types of cancers both in vitro and in vivo. Indeed, ICG-001 (or the structural derivative PRI-724) inhibits both tumor growth in murine colon cancer models that are Wnt dependent [17] and proliferation and migration in vitro and in vivo in gastric and pancreatic cancers [18,19]. On the other hand, the specificity of this inhibitor enables it to not interfere with others β-catenin-mediated cellular functions and therefore could have less toxicity in patients. The safety and toxicity of PRI-724 has been tested in Phase I clinical trials and is so far well tolerated by the patients (NCT02195440) [20], and new clinical trials are recruiting to determine the safety and tolerability of this inhibitor (NCT04688034, NCT04047160). Whatever, PRI-724 efficacy for cancer treatment has not been yet evaluated in clinical trials.

In our study, the effect of ICG-001 was evaluated in three human osteosarcoma cell lines, KHOS, MG63 and 143B. ICG-001 treatment strongly inhibits osteosarcoma cell proliferation through a cell cycle blockade in the G0/G1 phase, but surprisingly increases migration of the three cell lines in vitro. Despite a beneficial effect on cell proliferation in vitro, ICG-001 is unable to modulate tumor growth in a mouse model of osteosarcoma, but significantly increases the metastatic dissemination to lungs, confirming a pro-migratory role of ICG-001 on osteosarcoma cells.

## 2. Results

### 2.1. Osteosarcoma Cell Lines Expressing High β-Catenin Levels Are Sensitive to ICG-001

Based on the literature, the canonical Wnt signaling pathway plays a complex role in the development of osteosarcoma. Indeed, some data describe an oncogenic role of the canonical Wnt signaling pathway whereas some others highlight an anti-tumorigenic role of this pathway in osteosarcoma patients [11,12,15]. Thus, in order to provide recent data about the implication of the canonical Wnt signaling pathway in osteosarcoma, a published RNA-seq study (GSE99671) including 15 paired samples of patients (15 osteosarcomas and 15 healthy bones) was analyzed for β-catenin expression [21]. This analysis of paired samples highlighted that β-catenin mRNA level was significantly increased in osteosarcoma biopsies compared to healthy bones (Figure 1a). Consistently, gene set enrichment analysis (GSEA) of expression data obtained from the same cohort of patients revealed a Wnt signature in osteosarcoma samples as compared to healthy bone samples from the same patient (Figure 1b). β-catenin mRNA expression was also evaluated in six human osteosarcoma cell lines. As indicated in Figure 1c, high levels of β-catenin mRNA are observed in all tested osteosarcoma cells, excepted MG63 cells, as compared to human mesenchymal stem cells (MSCs). Western blot analyses confirmed that all osteosarcoma cell lines express β-catenin protein although an over-expression at the protein level compared to hMSC was not demonstrated in our culture conditions (Figure 1d). Similar results were observed both in the cytoplasmic and the nuclear compartments (data not shown). In addition, TCF/LEF luciferase gene reporter assay indicated that the Wnt/β-catenin signaling pathway is functional and can be activated by Wnt3a in KHOS, MG63 and 143B cells (Figure 2a). These results are mainly in favor of an aberrant activation of the canonical Wnt signaling pathway in osteosarcoma. Based on these data, the effects of ICG-001, a CBP/β-catenin inhibitor, were investigated in three human osteosarcoma cell lines, KHOS, MG63 and 143B.

After demonstrating that β-catenin protein level was unaffected by ICG-001 (Appendix A), the efficiency of this inhibitor was first evaluated by TCF/LEF luciferase gene reporter assay in KHOS, MG63 and 143B cells. ICG-001 dose-dependently decreased Wnt3a-induced TCF/LEF dependent luciferase activity in the three cell lines (Figure 2a). KHOS cells seemed to be the most sensitive cells to ICG-001 with a significant decrease of luciferase activity from 5 µM whereas a dose of 10 µM was necessary to observe a significant inhibition of luciferase activity in MG63 and 143B cells. Moreover, ICG-001 significantly diminished mRNA expression of two β-catenin target genes, AXIN2 and BIRC5 in KHOS and 143B cells with an inhibition higher than 50%. Only BIRC5 mRNA level was significantly decreased by 25% in MG63 cells that seem to be less sensitive to ICG-001, perhaps because of a lower β-catenin expression compared to KHOS and 143B cells (Figure 2b). Moreover, SURVIVIN protein level, encoded by BIRC5 gene, was decreased after 48 h of ICG-001 treatment in the three cell lines (Figure 2c). Taken together, these data revealed that ICG-001 inhibits the canonical Wnt signaling pathway in osteosarcoma cells. 

### 2.2. ICG-001 Induced a Cell Cycle Blockade in the G0/G1 Phase in Osteosarcoma Cells

To investigate the effects of β-catenin inhibition on osteosarcoma cell viability, cells were treated with increasing doses of ICG-001 for 24 h, 48 h and 72 h and viable cells were analyzed by Crystal Violet staining. ICG-001 decreased osteosarcoma cell viability in a time and dose dependent manner with IC50 of 0.83 µM, 1.05 µM and 1.24 µM at 72 h for KHOS, MG63 and 143B respectively (Figure 3a and Table 1). After 72 h of ICG-001 treatment, less than 20% of MG63 cells were still viable and less than 8% of KHOS and 143B as compared to untreated cells. To complete these results, cell death was analyzed after 72 h of ICG-001 treatment by Trypan Blue Counting. ICG-001 did not modify the proportion of dead cells, which remained between 4.1 and 6.3% whatever the treated cell line, suggesting that the decrease in viable cells under ICG-001 treatment was due to an inhibition of cell proliferation rather than an induction of cell death (Figure 3b). Moreover, a caspase-3 activity assay was performed, together with PARP cleavage analysis by western blot in KHOS cells. ICG-001 was not able to induce either PARP cleavage (Figure 3c) or caspase-3 activity (Figure 3d), confirming that ICG-001 treatment only induced an inhibition of cell proliferation without induction of apoptosis.

As evidenced by flow cytometry analysis of cell cycle, the decrease in osteosarcoma cell viability was associated with a cell cycle blockade in the G0/G1 phase. Indeed, ICG-001 significantly increased the proportion of KHOS, MG63 and 143B cells in the G0/G1 phase by 31.5%, 22.9% and 24.5% respectively. In parallel, a decrease in the proportion of cells in the S phase was observed from 38.5% to 10.4% for KHOS, from 31.1% to 7.3% for MG63 and from 26.3% to 8.9% for 143B cells after 24 h of ICG-001 treatment (Figure 4a). Then, in order to better understand the mechanisms responsible for this G0/G1 phase blockade, the expression of several proteins implicated in the cell cycle regulation was analyzed by western blot in KHOS cells. ICG-001 strongly inhibits CYCLIN D1 and CYCLIN D3 expression in KHOS cells, associated with a decreased Retinoblastoma protein (RB) phosphorylation, after 12 h and 24 h of treatment. In parallel, ICG-001 promotes P21WAF1 accumulation after 12 h and 24 h of ICG-001 treatment (Figure 4b). All these modulations repressed S phase genes expression and consequently induced a blockade of G1 to S phase transition (Figure 4c).

### 2.3. ICG-001 Stimulates Osteosarcoma Cell Migration

Given that the presence of metastasis is associated with a poor prognosis for osteosarcoma patients, cell migration was analyzed after ICG-001 treatment using both a transwell assay and a real time assay with the xCELLigence system. Osteosarcoma cells were treated during 24 h with 10 µM of ICG-001 before seeded in Boyden Chamber and cell counting of migrated cells was performed 8 h later. ICG-001 induced a significant 3-fold, 1.8-fold and 4-fold increase for KHOS, MG63 and 143B cell migration, respectively (Figure 5a). Results obtained with the xCELLigence system confirmed that ICG-001 increased KHOS and 143B osteosarcoma cell migration at least until 8 h after seeding (Figure 5b). Indeed, the cell index induced by ICG-001 treatment reached 1.22-fold increase for KHOS cells 8 h after seeding and 1.26-fold increase for 143B cells (Figure 5c). The migration of MG63 cells could not be evaluated with the xCELLigence system due to the very low migratory properties of this cell line. After 24 h, the Cell Index did not longer significantly differ between control and ICG-001-treated cells, suggesting that the inhibition of cell proliferation became predominant compared to the stimulation of cell migration.

In order to determine whether the enhanced migration was specific to the disruption of the CBP/β-catenin interaction, KHOS cell migration was evaluated after XAV-939 treatment, a tankyrase inhibitor that decreases β-catenin level [22]. Interestingly, XAV-939 decreased KHOS cell viability in a dose dependent manner as observed after ICG-001 treatment (Appendix A). However, 24 h of treatment with 10 µM of XAV-939 also drastically reduced KHOS cell migration (Appendix A). Thus, the migration of osteosarcoma cells seems to be solely induced by ICG-001.

### 2.4. ICG-001 Stimulates Metastatic Dissemination to Lungs in a Pre-Clinical Mouse Model of Osteosarcoma

ICG-001 inhibits osteosarcoma cell proliferation through a cell cycle blockade in the G0/G1 phase but stimulates the migration of osteosarcoma cells in vitro. Thus, to determine the predominant effect of ICG-001 in vivo, a mouse model of osteosarcoma xenograft was used. ICG-001 treatment, at a dose of 50 mg/kg/day, (diluted in DMSO) was initiated 7 days after para-tibial injection of KHOS cells in nude mice. First, it is important to note that DMSO did not impact either the tumor growth or the survival as compared to control group (Figure 6a,b). In the following series, ICG-001 treatment was only compared to DMSO treated mice for analyses. No significant differences of primary tumor volume could be observed between different groups, reaching, 34 days after KHOS cells injection, a mean of 1600 mm^3^ in ICG-001-treated mice versus 1380 mm^3^ in the DMSO group (Figure 6a). Moreover, Ki67 staining on osteosarcoma biopsy sections did not differ between ICG-001-treated group to DMSO-treated group confirming that ICG-001 seems not affect tumor growth in the pre-clinical model of osteosarcoma (Figure 6c). Consequently, mouse event free survival was not altered by ICG-001 treatment. Indeed, control mice were euthanized (tumor volume > 2500 mm^3^) from 38 to 39 days, DMSO treated mice from 39 to 52 days and ICG-001 treated mice from 35 to 47 days (Figure 6b).

Lung metastases were evaluated in this pre-clinical model of osteosarcoma by two complementary approaches: a macroscopic counting of metastases using binocular loupe and a histological counting on lung sections after Hematoxylin-Eosin staining. In accordance to in vitro results obtained on cell migration, ICG-001 significantly increased the metastatic dissemination to lungs in this pre-clinical model of osteosarcoma as evidenced by the two techniques. Lungs from DMSO-treated mice exhibited a mean of two macroscopic metastases at time of euthanasia whereas ICG-001 treated mice developed a mean of six macroscopic lung metastases. Moreover, one of the ICG-001 treated mice presented more than 60 macroscopic lung metastases and two other more than 10 metastases while no more than five metastases was reported for each DMSO-treated mouse (Figure 6d). To complete this analysis of metastases, a counting of metastatic nodules was performed on lung sections after Hematoxylin-Eosin staining. Thus, a mean of 20 metastatic nodules were counted in DMSO-treated mice and 80 nodules in ICG-001-treated mice with three evidencing higher than 180 metastatic nodules (Figure 6e). These pre-clinical results indicate that the predominant effect of ICG-001 in vivo seems to be the induction of the cell migration associated with the occurrence of lung metastases rather than an inhibition of primary tumor progression.

## 3. Discussion

The implication of the canonical Wnt/β-catenin signaling pathway in the development of osteosarcoma has been evaluated but data remain controversial. Indeed, some data described an aberrant activation of the canonical Wnt/β-catenin signaling pathway responsible for increased tumorigenicity and metastatic dissemination of osteosarcoma whereas some other studies highlighted an inactivation of this pathway in some osteosarcoma patients [13,14,15,16]. Our analysis of RNA seq data including 15 osteosarcoma patients and our screening of the level of β-catenin mRNA expression in six osteosarcoma cell lines (Figure 1) were consistent with a possible aberrant activation of the Wnt/β-catenin pathway in osteosarcoma. Thus, we proposed to block specifically the interaction between β-catenin and CBP using the drug ICG-001 and consequently to analyze the impact of the β-catenin/CBP dependent transcription regulation on primary and metastatic osteosarcoma development. 

Our present findings have established that ICG-001 decreased osteosarcoma cell proliferation in vitro through a G0/G1 cell cycle blockade (Figure 3 and Figure 4). These observations were consistent with most of previous studies evaluating the relevance of targeting the canonical Wnt pathway in osteosarcoma, using pharmacological or molecular approaches. As an example, the tankyrase inhibitor JW24 decreased β-catenin nuclear accumulation and reduced osteosarcoma cell growth [23]. A LRP5 dominant negative was also shown responsible for a reduced cell proliferation in vivo [24]. Interestingly, the implication of micro-RNAs (miR) has also been evaluated, such as miR-885-5p or miR-107 that decrease cell proliferation and migration through the inhibition of β-catenin. Other studies demonstrated that pharmacological inhibition or depletion of GSK3β by siRNA decreases osteosarcoma cell proliferation despite an increased nuclear translocation of β-catenin and expression of its target genes [25]. Likewise, treatment with the anti-Dickkopf 1 (DKK1, antagonist of the Wnt/β-catenin signaling pathway), BHQ880, reduces osteosarcoma development and lung metastasis in mouse models of patient-derived osteosarcoma xenograft [26], supporting a tumor suppressor role of β-catenin in osteosarcoma. 

The inhibition of proliferation has been related to a cell cycle blockade in the G0/G1 phase. ICG-001 decreased the expression of two cyclins, cyclin D1 and cyclin D3 as well as Rb phosphorylation while inducing p21 accumulation (Figure 4). Cyclins D1 and D3 and their partners CDK4/6 regulate S phase entry through Rb phosphorylation since hyperphosphorylation of Rb leads to activation of E2F transcription factors and the expression of their target genes responsible for cell cycle progression (Figure 4c) [27]. ICG-001, by reducing cyclins D1 and D3 levels, decreases Rb phosphorylation, leading to a blockade of G1 to S phase transition and consequently the progression of the cell cycle. The cyclin-dependent kinase inhibitor p21 also contributes to the cell cycle blockade in G0/G1, inhibiting Cyclin-Dependent Kinase 2 (CDK2), CDK3, CDK4 and CDK6-associated kinase activity (Figure 4c) [28,29]. Cyclin D1 is a direct target of the Wnt/β-catenin signaling pathway whereas the increased expression of cyclin D3 following β-catenin activation could be mediated through c-Jun, as demonstrated in myogenic cells [30]. Several studies also evidenced that p21 gene expression is negatively regulated by the β-catenin/TCF signaling and consequently that β-catenin inhibition leads to p21 accumulation in different cell lines including Human Embryonic Kidney (HEK) 293 cells, vascular smooth muscle cells and breast cancer cells [31,32,33]. Despite an anti-proliferative effect of ICG-001 in vitro, no effect could be observed in a pre-clinical mouse model of osteosarcoma. Other pro-tumoral factors are present in the tumor microenvironment, and may support tumor growth such as TGFβ or BMPs [34]. Thus, other signaling pathways could be activated in osteosarcoma cells and counterbalance cell cycle inhibition observed in the presence of ICG-001, such as activation of cyclin D1 gene transcription by Jun/Fos or STAT3 factors [35]. Another very important point concerns the bioavailability and biodistribution of ICG-001, as the highly hydrophobic molecule was injected into a mixture of DMSO and sesame oil intraperitoneally and its presence in the primary tumor could not be confirmed although a decrease in AXIN2 mRNA level was observed in the tumor (data not shown). 

In contrast to previous reports describing a pro-apoptotic role of β-catenin in osteosarcoma cells, ICG-001 treatment does not induce cell death in the three tested osteosarcoma cell lines. Indeed, SOST gene (encoding Sclerostin) silencing, leading to Wnt/β-catenin activation, down-regulates capsase-3 dependent apoptosis in osteosarcoma cells [36]. Therefore, the inhibition of the Wnt/β-catenin signaling pathway with tankyrase inhibitors, JW24 or IWR-1, promotes capase-3-mediated apoptosis in association with reduced cell proliferation in osteosarcoma [23,37]. However, one study has described that β-catenin knockdown by siRNA only slightly increases apoptosis in MG63 cells [38], suggesting a limited role of β-catenin in osteosarcoma apoptosis, as observed in our study. 

The most important discrepancy between our study and others dealing with the role of the Wnt/β-catenin signaling pathway in osteosarcoma development concerns its role in cell migration. Indeed, our study demonstrated that ICG-001 significantly promotes osteosarcoma cell migration in vitro and lung metastases establishment in a pre-clinical mouse model of osteosarcoma. Only one previous study analyzed the effects of a specific inhibitor of β-catenin/CBP interaction, PRI-724, a derivate of ICG-001 in osteosarcoma. The authors observed a decrease in cell proliferation in accordance with our results but highlighted an inhibition of osteosarcoma cell migration in vitro [39]. However, in that study, migration has been analyzed after 24 h of PRI-724 treatment with a dose 2-fold more elevated that the dose used in our study. Thus, we can hypothesize that the authors used a too high dose to observe specific effects on cell migration and/or that treatment was too long to observe only the effects of PRI-724 on cell migration without the reduced proliferation affecting the results. Indeed, our real time migration assay indicated that after 24 h of treatment, we did not observe an enhanced migratory effect of ICG-001 anymore, or even a low decrease in cell migration in 143B cells, probably due to a predominant effect on cell proliferation rather than on cell migration (Figure 5b). Moreover, we confirmed that ICG-001 promotes cell migration in our mouse pre-clinical model of osteosarcoma as we observed a significant increase in lung metastatic dissemination (Figure 6). Interestingly, a recent study is in accordance with our results, describing that Secreted Protein Acidic and Rich in Cysteine Like-1 (SPARCL1) protein suppresses osteosarcoma metastasis through activation of the Wnt/β-catenin signaling pathway [40].

Our results sustain that the role of the Wnt/β-catenin signaling pathway in osteosarcoma progression is highly complex since we highlighted a beneficial effect of ICG-001 on osteosarcoma cell proliferation on the one hand, but, a pro-metastatic role of this β-catenin inhibitor, on the other hand. To our knowledge, it is the first description of such a “dual effect” of a Wnt/β-catenin inhibitor on osteosarcoma development. Indeed, previous studies described either an oncogenic or a tumor suppressive role of β-catenin in osteosarcoma development and lung metastasis [13,14,15,16] but no studies have shown a pro-proliferative and anti-metastatic role of the Wnt/β-catenin signaling pathway in osteosarcoma. Moreover, it is important to consider the heterogeneity of osteosarcomas during analyses about the role of the Wnt/β-catenin signaling pathway in osteosarcoma cell lines. Indeed, MG-63 cells express low β-catenin mRNA and protein levels compared to KHOS and 143B cells. Nevertheless, the Wnt/β-catenin pathway can be activated by Wnt3a in MG63 cells (Figure 2a). Nevertheless, a strong inhibition of Survivin protein level was observed after ICG-001 treatment and the MG63 cells remain sensitive to ICG-001-induced decreased proliferation. Consequently, the basal level of β-catenin in osteosarcoma does not seem sufficient to predict sensitivity of osteosarcoma cells to ICG-001 and perhaps other β-catenin inhibitors. 

Finally, although ICG-001 decreases cancer cell migration in several cancer cells and pre-clinical models of tumors, its role in osteosarcoma seems to be in favor of a pro-metastatic effect. Consequently, it should be necessary to determine whether this deleterious effect is restricted to osteosarcoma or whether ICG-001 could enhance metastatic dissemination in other types of cancers. Currently, clinical trials allow to conclude that PRI-724 is well tolerated by the patients (NCT02195440) [20] but its efficacy in cancer treatment has not been evaluated. Based on our results, the potential beneficial effect of this Wnt/β-catenin inhibitor or its derivative PRI-724 should be analyzed in vivo before evaluating their effects in clinical trials including cancer patients.

To conclude, our study seems to add another level of complexity regarding the role of the Wnt/β-catenin signaling pathway in osteosarcoma pathogenesis, suggesting that the effect of β-catenin in osteosarcoma development could depend on the transcriptional co-activators with which it is associated to regulate its target genes expression. 

## 4. Materials and Methods

### 4.1. Cell Culture and Chemicals Reagents

KHOS (ATCC-CRL 1544), MG63 (ATCC-CRL 1427) and 143B (ATCC-CRL 8303) human osteosarcoma cell lines were cultured in Dulbecco’s Modified Eagle’s Medium (DMEM, Lonza) supplemented with 10% Fetal Bovine Serum (FBS, Biowest). Wnt3a was diluted in Dulbecco’s Phosphate Buffer Saline (DPBS) with 0.1% Bovine Serum Albumin (BSA). ICG-001 and XAV-939 (R&D Systems) were diluted in Dimethyl Sulfoxide (DMSO, Sigma), which was used as control vehicle (Vh) in each experiment.

### 4.2. RNA Seq Analysis

RNA seq data were downloaded from Gene expression Omnibus database (GSE99671, https://www.ncbi.nlm.nih.gov/geo/query/acc.cgi?acc=GSE99671 (accessed on 29 April 2019)). These published data allow the comparison of differential expression of β-catenin in Reads Per Kilobase Million (RPKM) between 15 human osteosarcoma tumors and their paired normal bone tissues [21]. Statistical analysis was performed with Graphpad Prism Software, using Wilcoxon matched-paired test, *p*-value = 0.0006. Gene Set enrichment analysis (GSEA) was also performed using GSEA software (http://software.broadinstitute.org/gsea/ (accessed on 29 April 2019)). Wnt target genes are from the Molecular Signature Database [41].

### 4.3. Luciferase Gene Reporter Assay

Cells were co-transfected with M50 Super 8x TOPFlash-Firefly luciferase (gift from Randall Moon, Addgene #12456, (TCF/LEF)8-lux) and phRLMLP-Renilla luciferase expression vector (Promega #AF025844) by using JetPEI (Polyplus-transfection). After transfection, cells were starved overnight (1% FBS), and treated with increasing concentrations of ICG-001 (1–10 µM) and 25 ng/mL of Wnt3a during 24 h in starved conditions. The Dual-Luciferase reporter assay system (Promega) was used to determine Luciferase activity. The Firefly luminescence was normalized to the Renilla luminescence and relativized by adjusting the average value of the DMSO/Wnt3a condition to 1. 

### 4.4. RT-qPCR

RNA from cell lines were extracted with NucleoSpin RNA plus kit (Macherey-Nagel). Then, 1 µg of RNA was retro-transcripted in cDNA using Maxima H Minus First Strand cDNA Synthesis Kit (Thermofischer). qPCR was performed on 20 ng cDNA using SYBR Select Master Mix (Life Technologies), using primers listed in the Table 2. Target gene expression was normalized to 18S, β2M and GAPDH levels in respective samples, and the comparative cycle threshold (Ct) method was used to calculated relative quantification of target mRNAs.

### 4.5. Western Blot

Cells were lysed in Radio-Immuno-Precipitation Assay (RIPA) buffer supplemented with protease inhibitor cocktail (Sigma), PMSF (phenylmethylsulforyl fluoride) and Sodium Orthovanadate. Protein concentration was determined using Bicinchoninic acid assay Kit (Sigma). Proteins (20 µg) were denatured by Laemli Buffer at 95 °C for 5 min, before to be loaded in Sodium Dodecyl Sulfate-Polyacrylamide gel. After electrophoresis separation, proteins were transferred to PolyVinyliDene Fluoride 0.45 µm membrane (Immobilon-FL, Merck Millipore). Membrane was saturated 1 h with Tris-Buffered Saline (TBS), 0.05%Tween (MM France), 3% Bovine Serum Albumin (BSA, Sigma), and was incubated overnight at 4 °C with primary antibodies in previous blocking buffer. Antibodies were used following the supplier recommendations: β-CATENIN (Cell Signaling Technology (CST) #2698), GAPDH (CST #2118), SURVIVIN (CST #2808), TUBULIN (CST #86298), PARP (CST #9542), VINCULIN (CST #13901), CYCLIN D1 (CST #2922), CYCLIN D3 (CST #2936), RB (CST #9309), PHOSPHO-RB Ser807/811 (CST #9308), P21 (CST #2947). After wash, membrane was saturated 15 min with TBS 0.05%Tween 5% non-fat milk, before incubation 1 h with secondary HRP coupled antibodies (Horseradish Peroxidase). The labelled proteins were detected by chemiluminescent detection (SuperSignal West Dura Extended Duration Substrate, ThermoFisher) using a Charged Coupled Device camera (GBox, Syngene).

### 4.6. Viability Assay (Crystal Violet Staining)

2000 KHOS and 143B cells and 3000 MG63 cells were seeded in 96-well plate. After adherence, cells were treated with increasing concentrations of ICG-001 (0.1–10 µM) or DMSO during 24 h, 48 h and 72 h. 2000 KHOS cells were also treated with increasing concentrations of XAV-939 (1–50 µM) or DMSO during 48 h. Cells were then fixed with 1% glutaraldehyde and stained by crystal violet. Finally, crystal violet staining was solubilized in Sorenson solution, and absorbance was measured at 570 nm with Victor² (Perkin Elmer).

### 4.7. Trypan Blue Counting

Cells were seeded in a 24-well plate, at 15,000 cells per well. After adherence, they were treated with ICG-001 10 μM or DMSO for 72 h. The supernatants were then collected, cells were washed with DPBS, which was also collected, trypsinized with Trypsine-EDTA (EthyleneDiamineTetraacetic Acid, Lonza) and harvested. Finally, cell suspensions containing live and dead cells, were counted after Trypan Blue staining.

### 4.8. Caspase-3 Activity

Adherent KHOS cells were treated with DMSO, ICG-001 or Staurosporine (STS, 1 μM as positive control) during 24 h and then lysed in RIPA buffer. Caspase-3 activity was assessed on 10 μL total cell lysates using the kit CaspACE Assay System, Fluorometric (Promega). Results were expressed in arbitrary units and corrected for protein content as determined by the Bicinchoninic acid assay Kit (Sigma).

### 4.9. Cell Cycle Analysis 

Adherent cells were treated during 24 h with DMSO or ICG-001 before collection of culture media. Then, cells were washed with DPBS that was collected, trypsinized, and fixed using 70% ethanol. Cells were washed with DPBS, centrifuged, and resuspended in a phospho-citrate buffer (Na_2_HPO_4_ 0.2 M, citric acid C_6_H_8_O_7_ 0.1 M, pH 7.5) for 30 min at room temperature. Cells were centrifuged and resuspended in DPBS, 0.12% Triton, 0.12 mM EDTA and 100 µg/mL RNAse A (Promega A797c), and incubated 30 min at 37 °C. Then, Propidium Iodide (5 µg/mL) was added for 20 min at 4 °C. Cell distribution was observed by flow cytometry in Platform Cytocell in Nantes (LSRII, Becton Dickinson). Data were analyzed with Flowlogic software.

### 4.10. Thymidine Synchronization 

Adherent cells were treated with 2 mM thymidine for 18 h, washed twice with DPBS and left in DMEM 10% FBS for 8 h. Then, cells were treated a second time with 2 mM Thymidine for 16 h and washed twice with DPBS before treatment with DMEM 10% FBS and DMSO or ICG-001 10 µM for 2, 8, 12 or 24 h. Cell distribution was observed by flow cytometry in Platform Cytocell in Nantes (LSRII, Becton Dickinson), after 2 h, 8 h, 12 h and 24 h or without any treatment. Data were analyzed with Flowlogic software. In addition, proteins were extracted for western blot analyses.

### 4.11. Migration Assay 

Cells were pre-treated with DMSO, 10 µM ICG-001 or 10 µM or 50 µM of XAV-939 for 24 h, before being seeded into a Boyden chamber with 8 µm pores size (50,000 cells/insert) and incubated for 8 h at 37 °C. An FBS gradient was performed in each Boyden chamber. Then, cells at the upper face were removed and cells at the lower face of the membrane were fixed with DPBS 1% Glutaraldehyde and stained with crystal violet. Migrated cells were counted using Image J software, in 14 fields by insert, and then represented by the average of cells per field per insert. In order to compare independent experiments, the number of DMSO treated-migratory cells was expressed as 100%.

### 4.12. Real-Time Migration Assay (xCELLigence^®^)

Cells were pre-treated with DMSO or 10 µM ICG-001 for 24 h, before being seeded into the upper well of a Cell Invasion Migration plate (60,000 cells/well, ACEA Biosciences, Inc.). Then, a FBS gradient was performed following the supplier’s recommendations. When cells migrate through the membrane with 8 µm pores and adhere to the microelectrodes on the underside of the membrane, they disrupt the current flow by increasing resistance. This parameter is monitored over time and allows the cell index (CI) to be calculated. For each experiment, the CI of the ICG-001 treated cells was relativized on the average of the CI of the DMSO-treated cells, the DMSO value being set to 1. 

### 4.13. Osteosarcoma Xenograft Murine Model

Four-weeks females NMRI-nude mice (Elevages Janvier) were bred under pathogen-free conditions in the Experimental Therapeutic Unit at the Medical School of Nantes, in agreement with the animal experimentation committee N°6 and referenced under the project number APAFIS #8405-2017010409498904_v3. The number of animals used to carry out this project was reduced to the strict minimum in order to conclude statistically. In a pre-established way, mice with weight loss higher than 10% were excluded from the study. The mice were anesthetized with an isoflurane/air mixture and 1 million KHOS cells was injected intra-muscularly close to the proximal tibia surface and then randomly assigned to the groups. After 1 week, when the tumor became palpable, the mice received a daily intraperitoneal injection of ICG-001 provided by CEISAM laboratory (Nantes) at 50 mg/kg/day (or DMSO). Tumor growth was monitored every 2 days with calipers and calculated according to the following formula: V = 0.5 × Longest side × (Shortest perpendicular side)^2^, without any blinding. The mice were euthanized when the tumor volume reached 2500 mm^3^.

### 4.14. Histological Analysis

At sacrifice, lungs were removed and fixed with 10% paraformaldehyde for 24 h before counting metastases with a binocular loupe. Finally, after embedding in paraffin, 3 µm sections were cut at four different levels spaced 200 µm each. The samples were then stained with Hematoxylin-Eosin, and metastases were counted after Nanozoomer acquisition. Paratibial tumors were collected at mass endpoint (1.5 cm^3^) and fixed with paraformaldehyde for 24 h. Tumors were embedded in paraffin, and sliced at 2 different levels, with a 3 µm thickness. Tumors slides were stained for Ki67 (Dako, M7240, 1:100), then incubated with an anti-mouse secondary antibody biotinylated (Dako, E0433, 1:400), and with the Streptavidin-Peroxidase (Dako, P0397, 1:800). The detection was done using DAB substrate and counterstained with Hematoxylin.

### 4.15. Statistical Analysis

Statistical analyses were performed using Graphpad Prism Software. For in vivo experiments, the comparison of tumor growth between groups was performed by a two-way ANOVA. The survival curves were analysed with a Log-rank (Mantel-Cox) test. Mann-Whitney test or unpaired t-test was performed to compare two groups. To compare more than 2 groups: ANOVA one-way was used when Brown-Forsythe showed variance equality, or Kruskal-Wallis in other cases. α = 0.05 and *p*-value ≤ 0.05 was considered statistically significant.

## Figures and Tables

**Figure 1 pharmaceuticals-14-00421-f001:**
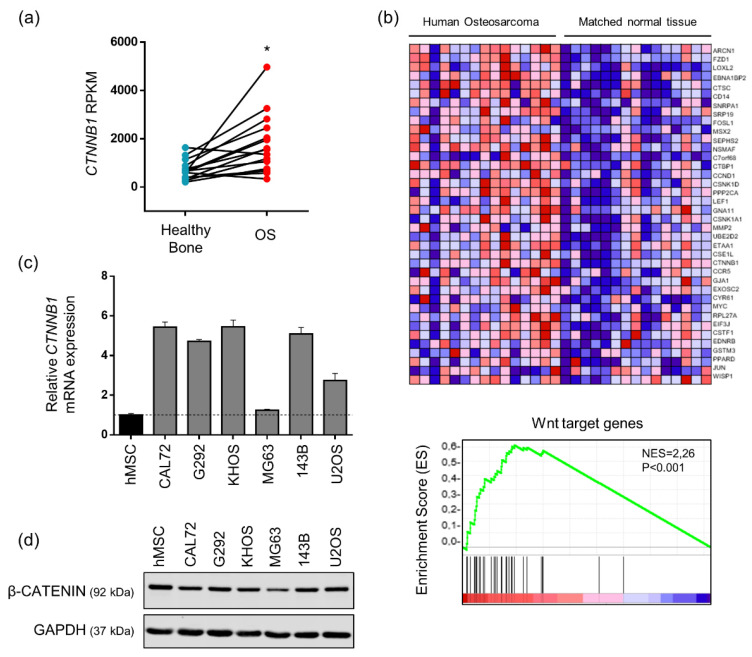
Osteosarcoma cells and tissues overexpressed β-catenin and its target genes. (**a**) Relative CTNNB1 gene expression in Reads Per Kilo base per Million (RPKM) in osteosarcoma (OS) samples and healthy bone tissues following bioinformatics analysis of RNAseq data GSE99671 (*n* = 15 paired patients). * *p* < 0.05. (**b**) Heatmap showing color-coded expression of Wnt target genes in osteosarcoma patients and healthy bone tissues from the same patients following bioinformatics analysis of RNAseq data GSE99671 (*n* = 15 paired patients). High expression is represented in red and low expression in blue (upper panel). GSEA plot showing the distribution of the Wnt target genes based on Gene Ontology (GO: 0090263) and KEGG pathway (M19428) databases and its enrichment score (ES). Color Scales are based on fold-changes of gene expression in osteosarcoma tissues compared to paired healthy bone tissues (lower panel) (**c**) Relative CTNNB1 gene expression in osteosarcoma cell lines compared to human Mesenchymal Stem Cells (hMSC). Bars indicate means ± SD of relative and normalized CTNNB1 mRNA expression determined by RT-qPCR (*n* = 1, performed in triplicate). (**d**) β-CATENIN protein level evaluated by western blot in osteosarcoma cell lines and hMSC (one representative experiment of three independent experiments).

**Figure 2 pharmaceuticals-14-00421-f002:**
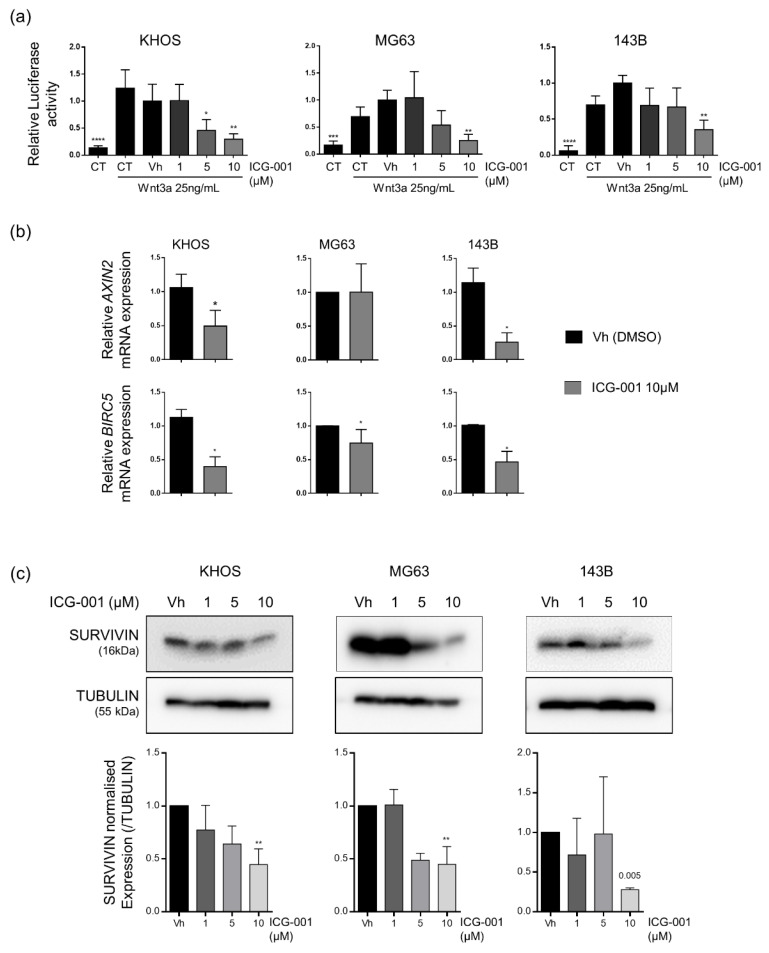
ICG-001 inhibits β-catenin dependent transcription in KHOS, MG63 and 143B osteosarcoma cell lines. (**a**) Relative Luciferase Firefly/Renilla activity ratio in cells co-transfected with TOPFLASH and pRL-TK constructs and treated or not (CT) with vehicle (Vh) or ICG-001 (1 to 10 µM). Wnt3a 25 ng/mL was added 1 h after ICG-001 treatment for a period of 24 h. Bars represent means ± SD of 5 independent experiments. * *p* < 0.05; ** *p* < 0.01; *** *p* < 0.001; **** *p* < 0.0001 vs. vehicle. (**b**) Relative and normalized AXIN2 or BIRC5 gene expression determined by RT-qPCR in cells treated with vehicle (Vh) or 10 µM of ICG-001 during 24 h (right panel). Bars represent means ± SD (*n* = 3, performed in duplicate). * *p* < 0.05 vs. vehicle. (**c**) SURVIVIN and TUBULIN protein level evaluated by western blot in cells treated with vehicle (Vh) or ICG-001 (1–10 µM) during 48 h (upper panel with one representative experiment of three independent experiments). Relative SURVIVIN protein quantification, normalized with TUBULIN protein expression (lower panel). Bars indicate means ± SD of three independent experiments. ** *p* < 0.01 vs. vehicle.

**Figure 3 pharmaceuticals-14-00421-f003:**
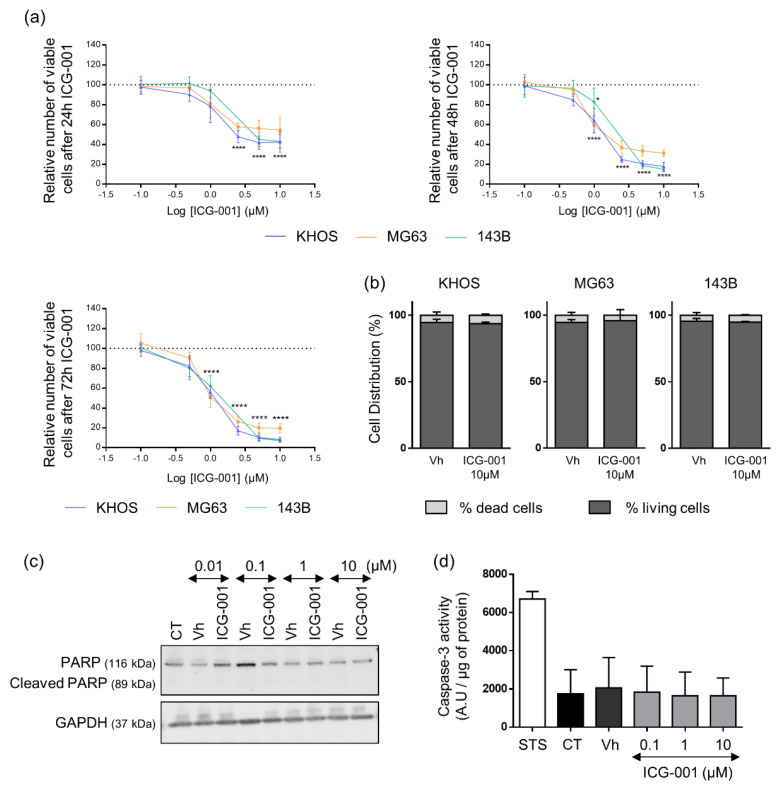
ICG-001 decreases KHOS, MG63 and 143B osteosarcoma cells viability. (**a**) Relative number of viable cells analyzed in cells treated with ICG-001 or vehicle (Vh) from 0.1 to 10 μM for 24, 48 or 72 h. At the end points, cells were stained with crystal violet and the curves represent the means ± SD relativized with the value of the vehicle (DMSO), of 5 independent experiments, each performed in triplicate. The dashed line represents the vehicle value set to 100. * *p* < 0.05; **** *p* < 0.0001 vs. vehicle. (**b**) Cell death evaluated after 72 h of ICG-001 treatment (10 µM). Live and dead cells were counted in Malassez chamber after Trypan Blue staining. Histograms represent means ± SD of three independent experiments, each performed in duplicate. (**c**) Representative image of PARP analyzed by western blot in KHOS cells treated with ICG-001 or vehicle (Vh) from 0.01 to 10 μM for 24 h. (**d**) Caspase-3 activity (expressed in Arbitrary Unit (A.U)/ μg of protein) determined on KHOS cell lysates after 24 h of ICG-001 or vehicle (Vh) treatment by a fluorometric assay. Staurosporine (STS, 1 μM) is used as positive control of caspase-3 activity. Histograms represent means ± SD of three independent experiments.

**Figure 4 pharmaceuticals-14-00421-f004:**
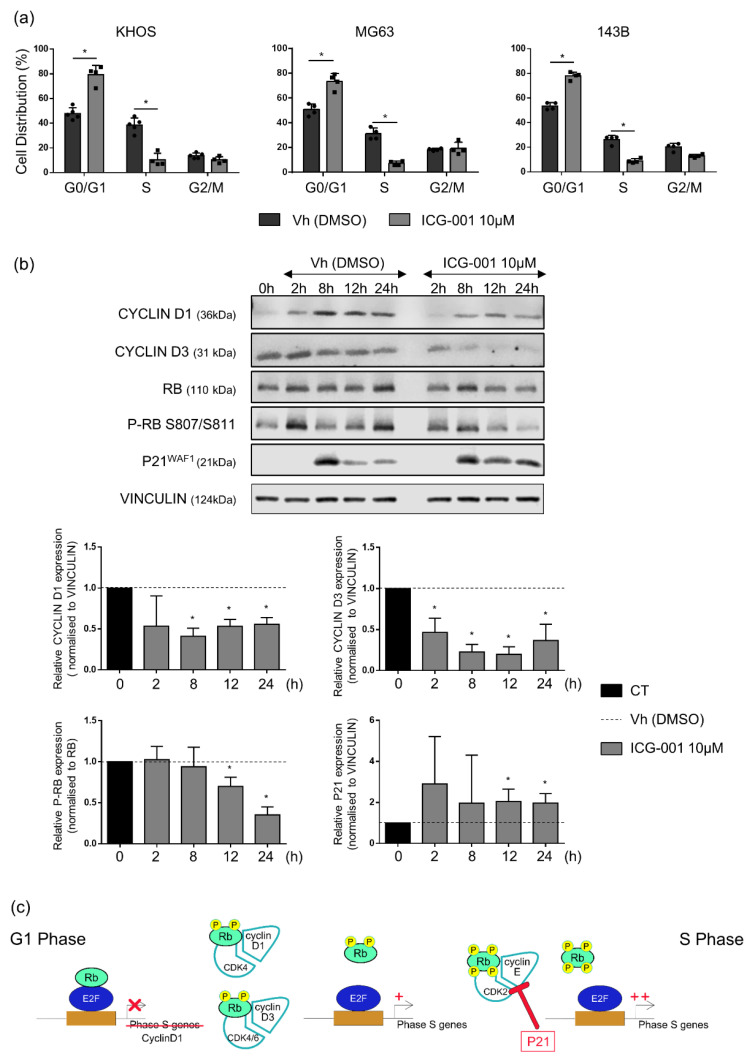
ICG-001 blocks KHOS, MG63 and 143B osteosarcoma cell lines in the G0/G1 phase. (**a**) Cell cycle analysis by flow cytometry in cells treated with ICG-001 (10 µM) or vehicle (Vh) for 24 h. Histograms represent means ± SD of three independent experiments. * *p* ≤ 0.05. (**b**) Cell cycle regulators protein expression in KHOS cells synchronized in early S phase with thymidine treatment, released in cell cycle during 0, 2, 8, 12 and 24 h, and treated with ICG-001 (10 µM) or vehicle (Vh). Representative images of three independent experiments. Histograms represent means ± SD of relative CYCLIN D1, CYCLIN D3 and P21 protein quantification relativized with the value of the vehicle (DMSO for each time) and normalised with VINCULIN and relative quantification of RB phosphorylated form protein expression relativized with the value of the vehicle (DMSO for each time) and normalized with total RB of three independent experiments. The dashed line represents the vehicle value set to 1 for each time. * *p* ≤ 0.05 vs. vehicle for each time. (**c**) Regulation of G1 to S phase transition implicating cyclins and cyclin dependent kinases (CDK) as well as Retinoblastoma protein (Rb) and p21.

**Figure 5 pharmaceuticals-14-00421-f005:**
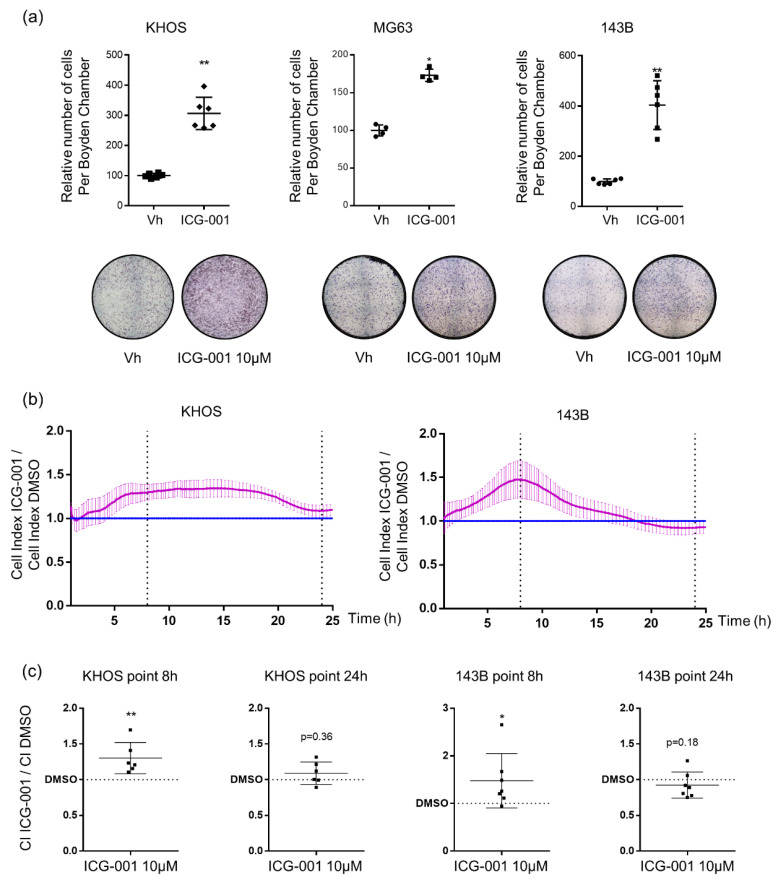
ICG-001 increases KHOS, MG63 and 143B osteosarcoma cell migration. (**a**) Migrated KHOS, MG63 and 143B cells counting after 24 h of ICG-001 (10 µM) or vehicle (Vh) pre-treatment and seeding in Boyden Chamber for 8 h. Bars represent means ± SD (KHOS *n* = 3; MG63 *n* = 2; 143B *n* = 3 each performed in duplicate). * *p* < 0.05; ** *p* < 0.01. (**b**) Real-time migration of KHOS and 143B cells after 24 h of ICG-001 (10 µM) or vehicle (Vh) and seeding in inserts of CIM plate. Curves represent means ± SEM of Cell Index (CI) of ICG-001 treated cells related to the CI of vehicle-treated cells, for three independent experiments, each performed in duplicate. (**c**) CI of ICG-001 treated cells relativized to CI of vehicle treated cells at 8 h and 24 h * *p* < 0.05; ** *p* < 0.01 vs. vehicle.

**Figure 6 pharmaceuticals-14-00421-f006:**
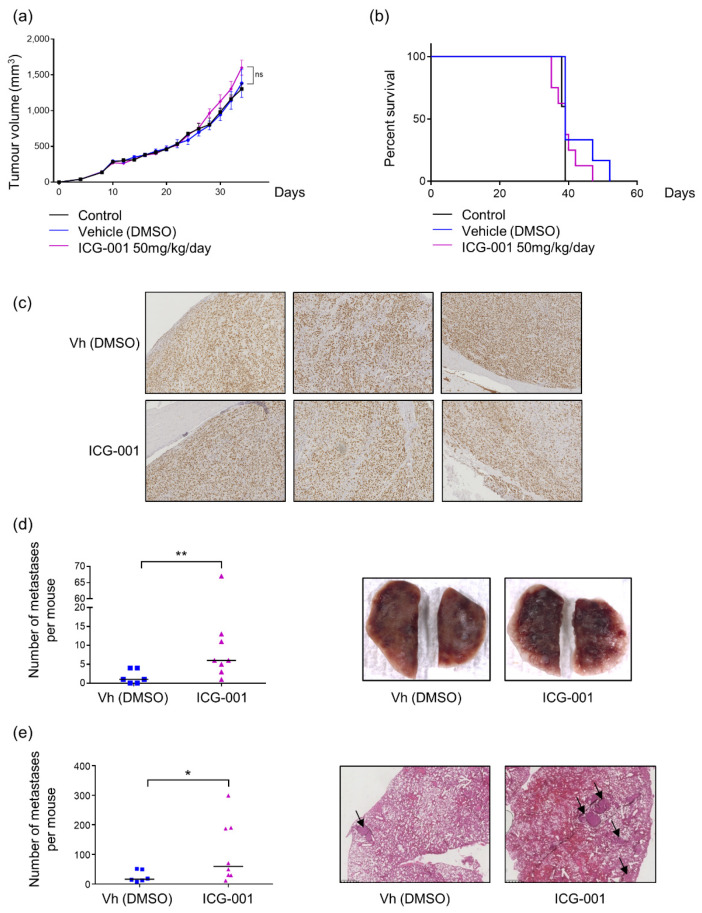
ICG-001 enhances metastatic dissemination to lungs in a mouse model of osteosarcoma. (**a**) Tumor growth monitoring by volume measurement every two days. Different curves (Control *n* = 5, DMSO *n* = 6, ICG-001 50 mg/kg/day *n* = 8) represent means ± SEM. (**b**) Survival curves, with a median survival of 39 days for each group. (**c**) Representative images of paratibial tumor sections slides from DMSO or ICG-001 mice groups stained with Ki67. Magnification × 20. (**d**) Macroscopic count of metastases, each dot represents the number of metastases per mouse, and bars represent medians. ** *p* ≤ 0.01. (**e**) Histological count of metastases after Hematoxylin-Eosin staining, each dot represents the number of metastases per mouse, bars represent medians. On the right panel, arrows indicate metastases * *p* ≤ 0.05.

**Table 1 pharmaceuticals-14-00421-t001:** Inhibitory Concentration 50 (IC_50_) of ICG-001 in KHOS, MG63 and 143B cells.

Cell Line	MG63	KHOS	143B
IC_50_ (24 h)	1.11 µM	1.27 µM	1.12 µM
IC_50_ (48 h)	0.87 µM	1.11 µM	1.59 µM
IC_50_ (72 h)	0.83 µM	1.05 µM	1.24 µM

**Table 2 pharmaceuticals-14-00421-t002:** Primers used for qPCR experiments.

Genes	Primers Sequences
*h18S*	Qf: cgattggatggtttagtgagg
	Qr: agttcgaccgtcttctcagc
*hAXIN2*	Qf: atgattccatgtccatgacg
	Qr: cttcacactgcgatgcattt
*h*β*2M*	Qf: ttctggcctggaggctatc
	Qr: tcaggaaatttgactttccattc
*hBIRC5*	Qf: aggaccaccgcatctctacat
	Qr: aagtctggctcgttctcagtg
*hGAPDH*	Qf: tgggtgtgaaccatgagaagtatg
	Qr: ggtgcaggaggcattgct

## Data Availability

RNA seq data were downloaded from Gene expression Omnibus database (GSE99671, https://www.ncbi.nlm.nih.gov/geo/query/acc.cgi?acc=GSE99671, accessed on 29 April 2019). The data presented in this study are available in the article.

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
