# Peer review of "ICG-001, an Inhibitor of the β-Catenin and cAMP Response Element-Binding Protein Dependent Gene Transcription, Decreases Proliferation but Enhances Migration of Osteosarcoma Cells"

_pharmaceuticals, 2021, doi:10.3390/ph14050421_

Round 1
Reviewer 1 Report
I have the following comments :
- The authors have not provided enough clinical background of the drug ICG-001 in terms of its efficacy and clinical use in osteosarcoma and other cancer types in the introduction section. This is crucial to understand the clinical relevance of this study.
- The effect of ICG-001 on the protein levels of beta-catenin has not been shown in the manuscript.
- The quality of western blots must be improved (especially 2c and 4b) since the data cannot be considered conclusive.
- The authors must describe the clinical relevance of this study in osteosarcoma patients in the discussion section.
- Is the increased migratory effect observed with ICG-001 just specific to this drug due to its particular chemical structure or all B-catenin inhibitors ? Have studies performed with other inhibitors of this class observed the same phenomenon?
Reviewer 2 Report
The authors tested both in vitro, using 3 osteosarcoma cell lines and in vivi, sing a xenograft mouse model, the ICG-001 inhibitor. They report that while the inhibitor efficiently reduces cell proliferation, it increases the ability of cells to migrate and metastasize in vitro and in vivo.
The findings are well supported by the data, I have only one request:
if possible, it would be nice to stain the mouse xenograft tumors for a proliferation marker, such as Ki67.
Round 2
Reviewer 1 Report
My comments have been addressed.